# The Effects of Volatile Anesthetics on Renal Sympathetic and Phrenic Nerve Activity during Acute Intermittent Hypoxia in Rats

**DOI:** 10.3390/biomedicines12040910

**Published:** 2024-04-19

**Authors:** Josip Krnić, Katarina Madirazza, Renata Pecotić, Benjamin Benzon, Mladen Carev, Zoran Đogaš

**Affiliations:** 1Department of Emergency Medicine, University Hospital of Split, Spinčićeva 1, 21000 Split, Croatia; 2Department of Anesthesiology, Reanimatology and Intensive Care Medicine, University Hospital of Split, Spinčićeva 1, 21000 Split, Croatia; 3Department of Neuroscience, University of Split School of Medicine, Šoltanska 2A, 21000 Split, Croatia; 4Department of Anatomy, University of Split School of Medicine, Šoltanska 2A, 21000 Split, Croatia

**Keywords:** acute intermittent hypoxia, isoflurane, sevoflurane, renal sympathetic nerve activity, phrenic nerve activity, obstructive sleep apnea, rats

## Abstract

Coordinated activation of sympathetic and respiratory nervous systems is crucial in responses to noxious stimuli such as intermittent hypoxia. Acute intermittent hypoxia (AIH) is a valuable model for studying obstructive sleep apnea (OSA) pathophysiology, and stimulation of breathing during AIH is known to elicit long-term changes in respiratory and sympathetic functions. The aim of this study was to record the renal sympathetic nerve activity (RSNA) and phrenic nerve activity (PNA) during the AIH protocol in rats exposed to monoanesthesia with sevoflurane or isoflurane. Adult male Sprague-Dawley rats (*n* = 24; weight: 280–360 g) were selected and randomly divided into three groups: two experimental groups (sevoflurane group, *n* = 6; isoflurane group, *n* = 6) and a control group (urethane group, *n* = 12). The AIH protocol was identical in all studied groups and consisted in delivering five 3 min-long hypoxic episodes (fraction of inspired oxygen, FiO_2_ = 0.09), separated by 3 min recovery intervals at FiO_2_ = 0.5. Volatile anesthetics, isoflurane and sevoflurane, blunted the RSNA response to AIH in comparison to urethane anesthesia. Additionally, the PNA response to acute intermittent hypoxia was preserved, indicating that the respiratory system might be more robust than the sympathetic system response during exposure to acute intermittent hypoxia.

## 1. Introduction

Respiratory modulation of sympathetic nervous activity is a well-described phenomenon, and synchronization of both systems is considered an important mechanism in maintaining optimal tissue perfusion and oxygen delivery [1,2]. Exposure to noxious stimuli, such as hypoxia and hypercapnia, leads to activation of the respiratory and sympathetic neurons in the brainstem and may strengthen the coordination of respiration and autonomic activity [3,4,5,6]. Intermittent hypoxia is a valuable model for studying obstructive sleep apnea (OSA) pathophysiology and implies repeated brief exposures to reduced oxygen levels interrupted by reoxygenation periods and, as such, has already been a subject of numerous studies [3,4,7,8,9,10,11,12,13,14]. Previous findings indicate that chronic intermittent hypoxia (CIH), a common feature of OSA, is associated with sustained increases in sympathetic nerve activity and arterial blood pressure [8,12]. Moreover, acute intermittent hypoxia (AIH) is an experimental intervention for initiating and enhancing long-term facilitation (LTF) of both respiratory and sympathetic systems [4,11,15], depending on the genetic background, animal species, anesthesia and the peripheral nerve recorded [16,17]. However, the evidence about the acute effects of brief, repetitive episodes of severe hypoxia on discharge patterns of renal sympathetic and phrenic nerve activity in animals anesthetized with volatile agents is scarce.

Sevoflurane and isoflurane are the most commonly used volatile anesthetics in everyday clinical practice, and many studies have proved their safety and effectiveness [18,19,20]. However, previous research has demonstrated that volatile anesthetic agents change breathing patterns, even in subanesthetic doses, during the acute ventilatory response to hypoxia [21,22,23,24,25]. Consequently, volatile anesthetics blunt the acute hypoxic ventilatory response in a dose-dependent manner, where the relative order of potency has been proved to be sevoflurane < isoflurane < halothane [22,23,26].

Volatile anesthetics alter the autonomic responses and blood pressure homeostasis mechanisms in a dose-dependent manner, either through a direct effect on vasculature and the heart or by an indirect effect on autonomic activity [27,28,29,30]. Although the global effect of volatile anesthesia is depression of autonomic activity, volatile anesthetics have been shown to either increase or decrease the sympathetic activity in animal models [31,32,33,34,35,36,37,38]. Such contrasting results likely depend on numerous factors, including the anesthetic and animal species used and the experimental setting but also differential and independent effects on sympathetic outflow to various target organs [37]. Considering the profound effects of volatile anesthesia on both respiratory and sympathetic functions and the synergistic interaction between the two systems, it remains unclear whether both systems are interdependently modulated during exposure to noxious stimuli during anesthesia with volatile anesthetics.

Previous studies have centered on AIH’s effect on long-term changes in the respiratory system. Still, they did not evaluate the acute effects on phrenic or sympathetic activity during hypoxic episodes under volatile agent monoanesthesia. Therefore, this study assessed the impact of rapid cyclic bouts of hypoxia during the AIH protocol on both renal sympathetic nerve activity (RSNA) and phrenic nerve activity (PNA) under monoanesthesia with sevoflurane and isoflurane while simultaneously recording both nerves. The aim of this study was to assess which system may be more resilient to repetitive noxae, and we hypothesized that the respiratory and sympathetic systems will be differentially modulated during exposure to acute intermittent hypoxia.

## 2. Materials and Methods

The procedures and protocols for this study were approved by the Ethical Committee for Biomedical Research of the University of Split School of Medicine (Split, Croatia) and the National Ethics Committee of the Veterinary Directorate, Ministry of Agriculture, Republic of Croatia. The animals were sourced from the University of Split School of Medicine Animal Facility, where they were maintained on a 12 h light/12 h dark cycle with free access to food and water prior to random allocation to the experimental groups for this study.

### 2.1. General Procedures

A total of twenty-four adult male Sprague-Dawley rats, weighing from 280 to 360 g, were selected and randomly divided into three groups: two experimental groups (sevoflurane group, *n* = 6; isoflurane group, *n* = 6) and a control group (urethane group, *n* = 12; Figure 1).

The animals in the control group were anesthetized with 20% urethane administered intraperitoneally (1.2 g/kg initially and a supplemental dose of 0.2 g/kg, if necessary). For each experiment in both experimental groups, one rat was placed in a transparent glass anesthesia box (in-house design) of 4 L volume and was exposed to a single volatile agent—sevoflurane or isoflurane. Fresh gas was obtained from a cylinder with a fixed mixture of 50% N_2_ and 50% O_2_ (fraction of inhaled oxygen; FiO_2_ = 0.5) connected to a calibrated vaporizer (isoflurane vaporizer 19.3, sevoflurane vaporizer 19.3; Dräger, Lubeck, Germany). In accordance with the previous findings, MAC values were determined at 2.4% for sevoflurane and 1.5% for isoflurane [22,23,39,40]. The induction of anesthesia was obtained with 3 MAC during the first 3 min and subsequently decreased to 2 MAC for the next 10–20 min. Spontaneous breathing was continuously monitored until adequate anesthesia depth was attained. The animal was then transferred to a surgical table, and the ventilation was maintained by means of a face mask (flow of 1 L/min).

The complete absence of hind paw withdrawal and corneal reflexes was used to confirm the adequacy of anesthesia before commencing any surgical procedures and frequently throughout each experiment. First, the cannulation of the trachea was performed through a midline ventral neck incision. Following the placement of the cannula, the animals were mechanically ventilated using a small animal ventilator (SAR 830-P; CWE Inc., Ardmore, PA, USA), and bilateral vagotomy was performed. The initial ventilatory settings were set to the following: respiratory rate of 50 breaths/min, inspiratory time of 0.6 s and positive end-expiratory pressure of 2–3 cm of H_2_O. The ventilator settings were adjusted depending on arterial blood gas analysis results (RAPIDPoint 500; Siemens Healthcare Limited, Surrey, UK). Both femoral veins and arteries were cannulated for continuous intravenous infusion of saline (1.5–2 mL/kg/h), blood sampling, continuous blood pressure monitoring and possible drug delivery.

Next, the animals were placed in a prone position and fixed in a stereotaxic frame (Lab Standard; Stoelting Co., Wood Dale, IL, USA). A dorsal approach at the level of C5 nerve rootlet was used to access the right phrenic nerve, and the retroperitoneal approach was used to access the left renal sympathetic nerve, as previously described [23,41]. Each intact nerve was mounted on a silver wire electrode and covered with silicone gel to prevent it from drying and minimize noise artefacts.

Following the completion of the surgical procedures, the vaporizer inspiratory concentrations for both anesthetics were set to 1.4 MAC, since that level was defined as the lowest MAC for providing adequate sedation in rats in previous studies [22,23,39,40]. Approximately thirty minutes were allowed to pass in order to obtain stable nerve signals and adequate level of anesthesia. The depth of anesthesia remained unchanged until the end of protocol, and gas mixture was changed only in the AIH protocol, more precisely during five hypoxic episodes. Throughout the experiment, the animals’ body temperatures were maintained at 37.0–38.5 °C by means of an external heating pad (FST; Heidelberg, Germany). After the experiment was completed, the animals were euthanized by an intravenous bolus of saturated potassium chloride solution.

### 2.2. Recording Parameters

The raw signals were amplified, filtered (300 Hz–10 kHz, band-pass filter), rectified, integrated (MA-1000 PowerLab Moving Averager module for System 1000 with a time constant of 50 ms; CWE Inc., Ardmore, PA, USA) and sampled at 20 kHz. Arterial blood pressure, RSNA and PNA signals were recorded simultaneously using Chart 5.4.2 for Windows software (ADInstruments, Bella Vista, Australia).

### 2.3. Acute Intermittent Hypoxia Protocol

Before exposure to the AIH protocol, the animals were ventilated with hyperoxic mixture (FiO_2_ = 0.5), and baseline renal sympathetic and phrenic nerve activities were established. The AIH protocol was identical in all studied groups and consisted in delivering 5 episodes of hypoxia at FiO_2_ = 0.09 for the duration of 3 min, separated by 3 min recovery intervals at FiO_2_ = 0.5. After the fifth hypoxic episode, the oxygen level was restored and maintained at baseline level until the end of the experiment. Arterial blood samples were drawn at baseline point (T0) and at 15 min following the end of the last hypoxic episode (T15).

### 2.4. Data Analysis

The recorded variables were analyzed during 20 s intervals at each experimental time point (before first hypoxia, i.e., baseline activity—T0; during each of five hypoxic episodes—TH1 to TH5; and at 15 min after the last hypoxic episode—T15). RSNA was analyzed and quantified from the integrated signal by measurement of the area under the curve (integral from minimum), and PNA was analyzed as an averaged PNA amplitude value at each time point using Chart 5.4.2. for Windows (AD Instruments, Bella Vista, Australia). Both nerve activities were expressed as fold change from the baseline activity. MAP was measured during the same time intervals and was expressed in the mmHg ± standard error of the mean. Arterial blood gas analysis (pH, PaO_2_ and PaCO_2_) was performed at T0 and T15.

### 2.5. Statistical Analysis

Statistical analyses were performed using GraphPad Prism version 10.0.0 (GraphPad Software, Boston, MA, USA) and MedCalc version 11.5.1.0 (MedCalc Software, Mariakerke, Belgium). The sample size was estimated using MedCalc and data obtained in our previous study [23]. With type I error of 0.05 and statistical power of 90%, it was determined that a minimum of 10 animals is required in the control group and 5 animals in the experimental groups. The nerve activity was normalized to conditions at baseline. Nerve activity was compared to the baseline by single-sample Wilcoxon test for each anesthetic. Response from 2nd to 5th hypoxic episode was modeled by linear regression. Differences between groups and baseline are expressed as fold change relative to the baseline. For comparison of MAP, pH, PaO_2_ and PaCO_2_ between experimental time points for each group, repeated measures ANOVA was used. One-way ANOVA was used for comparison of MAP at baseline between groups. *p* values less than 0.05 were considered significant.

## 3. Results

### 3.1. Overall Activity of Renal Sympathetic and Phrenic Nerves

RSNA and PNA were recorded at baseline, through five episodes of hypoxia and after 15 min of recovery (Figure 2).

Descriptively, PNA showed an increase in activity during the first episode of hypoxia, which was followed by a plateau until the recovery phase (Figure 3b). On the other hand, RSNA only showed a clear increase in the urethane group, and the following four episodes of hypoxia were characterized by a plateau in volatile anesthetic groups, whereas the urethane group showed a steady decline (Figure 3a).

### 3.2. Activity of RSNA and PNA at the Baseline and during the First Hypoxic Challenge

RSNA demonstrated a median 1.61-fold increase (95% CI 1.08 to 1.89; *p* = 0.009) when compared to baseline conditions in rats that were anesthetized with urethane (Figure 4a). On the other hand, rats anesthetized with sevoflurane or isoflurane demonstrated a median increase in activity of 1.17 (95% CI 0.87 to 1.17; *p* = 0.115) and 0.89-fold (95% CI 0.87 to 1.16; *p* = 0.345), respectively, when compared to the baseline (Figure 4a).

When compared to the baseline, PNA was increased by a median of 4.27-fold (95% CI 2.78 to 7.38; *p* < 0.001) in urethane-anesthetized rats, followed by rats anesthetized by isoflurane (median fold change = 3.9; 95% CI 2.58 to 25.3; *p* = 0.027) and rats anesthetized with sevoflurane (median fold change = 3.45; 95% CI 1.93 to 7.55; *p* = 0.027; Figure 4b).

### 3.3. Activity of RSNA and PNA during Subsequent Episodes of Hypoxia and Recovery

Responses from the second to fifth episodes hypoxia were modeled with linear regression in each rat, and slopes of the model are used for further comparisons.

If dynamics of response from the second to fifth episodes hypoxia are analyzed in RSNA, no conclusive trend can be found, i.e., slopes of all groups are centered around zero (Figure 5a). Although rats anesthetized with urethane descriptively show a decaying dynamic (Figure 5a), it cannot reach the significance threshold probably due to large data spread. On the other hand, PNA nerve activity decays in all three groups of rats, with the decay being fastest in the urethane group (−0.82-fold change per hypoxia episode; 95% CI −1.79 to −0.37; *p* = 0.002), followed by isoflurane-anesthetized rats (−0.7-fold change per hypoxia episode; 95% CI −4.8 to −0.3, *p* = 0.027) and the sevoflurane group (−0.37-fold change per hypoxia episode; 95% CI −0.84 to −0.17; *p* = 0.043; Figure 5b).

When it comes to comparison of both RSNA and PNA between the last hypoxic episode and T15, no significant differences were observed.

### 3.4. Mean Arterial Pressure and Acid–Base Status

At baseline, mean arterial pressure (MAP) was significantly lower in the isoflurane-anesthetized group in comparison to the urethane-anesthetized group (82.1 ± 5.0 mmHg vs. 98.5 ± 3.7 mmHg, *p* = 0.033; Table 1).

In the sevoflurane group, no significant changes in MAP were observed during exposure to the AIH protocol. However, exposure to AIH produced significant decreases in MAP during TH3 and TH4 (TH3: 61.5 ± 6.1 mmHg, *p* = 0.020; TH4: 60.5 ± 6.4 mmHg, *p* = 0.038) in comparison to the baseline (T0: 82.1 ± 5.0 mmHg; Table 1) in animals under isoflurane anesthesia. In urethane-anesthetized animals, significant decreases in MAP were observed during TH3-TH5 (TH3: 77.2 ± 5.2 mmHg, *p* = 0.046; TH4: 72.9 ± 5.1 mmHg, *p* = 0.013; TH5: 72.9 ± 5.6 mmHg, *p* = 0.021) in comparison to the baseline (T0: 98.5 ± 3.7 mmHg; Table 1). At T15, no significant differences in MAP were observed in comparison to the baseline values for all studied groups.

No significant differences in pH and the partial pressure of oxygen in arterial blood (PaO_2_) were observed between the baseline and T15 time point in each studied group (Table 2). The partial pressure of carbon dioxide in arterial blood (PaCO_2_) value was lower at T15 in comparison to the baseline in the urethane-anesthetized group (52.0 ± 1.7 mmHg vs. 48.3 ± 2.0 mmHg, *p* = 0.045, Table 2), whereas in isoflurane and sevoflurane no significant changes in PaCO_2_ were observed.

## 4. Discussion

This study examined the effects of volatile anesthetics on sympathetic and respiratory activity in rats exposed to rapid cyclic bouts of severe hypoxia during AIH protocol while concomitantly recording the RSNA and PNA. The main finding of the present study was that volatile agents, isoflurane and sevoflurane, blunted the RSNA response to the first hypoxic challenge during exposure to the AIH protocol. Moreover, in subsequent episodes of severe hypoxia, the sympathetic response remained suppressed even in a hyperoxic background at an equivalent minimal alveolar concentration (MAC). As for the respiratory activity, monoanesthesia with volatile agents evoked a significant increase in PNA during the first severe hypoxic challenge, and the response was preserved throughout the subsequent hypoxic episodes. These results indicate that the respiratory system might be more robust than the sympathetic system response during exposure to acute intermittent hypoxia.

Exposure to intermittent hypoxic stimuli is a well-established model to study the effects of repeated hypoxic events on various organ systems, replicating the occurrence of repetitive desaturation episodes present in sleep-disordered breathing [42,43]. Acute exposures to intermittent hypoxia induce increased respiratory and sympathetic effort and may lead to the development of long-term respiratory and sympathetic plasticity [3,4,11,22,23,44]. Previous studies have demonstrated that the sympathetic plasticity evoked by AIH results from synergistic actions between respiratory and sympathetic control centers but can also be independently elicited through the elevation of the sympathetic tone by AIH [4,45,46,47]. However, the interaction between respiratory and sympathetic systems during episodic exposures to severe hypoxia remains unclear. Our experiments raise the possibility that the response of the respiratory system to severe hypoxic challenges in the hyperoxic background might be more robust. In contrast, volatile anesthetics may modulate the robustness of the sympathetic response to AIH. Indeed, our previous study provided evidence that volatile anesthetics preserved the respiratory response measured by PNA during exposure to severe intermittent hypoxic stimuli at equivalent MACs of isoflurane and sevoflurane [23].

The studies that evaluated the effect of volatile anesthetics on sympathetic nerve activity yielded different results. Except for halothane, which can markedly suppress the RSNA, other volatile anesthetics, such as enflurane, isoflurane and desflurane, generally demonstrate a biphasic pattern that is characterized by sympathoexcitation at inhaled concentrations of <1.5 MAC and sympathodepression at higher concentrations, i.e., >2.5 MACs [31,34,35,36]. On the contrary, sevoflurane produced either a sympathoexcitatory effect or no changes in the RSNA [32,33,37,38]. Although the impact of volatile anesthetics on sympathetic and respiratory systems has been extensively studied before [28,29,30], the sympathetic system response to AIH in animals anesthetized with volatile anesthetics has yet to be completely elucidated. In the present study, both isoflurane and sevoflurane blunted the RSNA response to severe intermittent hypoxic stimuli at equivalent MACs. The finding that the renal sympathetic activity is blunted during the course of exposure to AIH under volatile anesthesia might also have clinical implications. RSNA has been proved to be important in the perioperative period and is known to be independently affected by volatile anesthetics [48]. Thus far, most studies have demonstrated an increase in RSNA in animals anesthetized with volatile anesthetics [31,37,48], whereas our study indicates the RSNA was blunted by volatile anesthetics, albeit in the environmental setting of AIH. The changes in the renal blood flow evoked by modifications in RSNA can have a profound impact on kidney excretory function and oxygenation and could contribute to the development of acute kidney injury in the perioperative setting [48]. Thus, such findings might influence the choice of an anesthetic for certain patient groups and types of surgery, but further research is necessary to assess the actual clinical relevance of these findings.

It is well established that the coupling of the respiratory and sympathetic systems varies according to the peripheral nerve recorded, animal species and experimental model applied [49]. Considering the reciprocal connections between neuronal networks within the ventrolateral medulla that generate the respiratory and sympathetic outflows, it is possible that noxious stimuli, such as severe hypoxia, differentially modulate the two systems under volatile anesthesia. Our findings support that rats anesthetized with volatile anesthetics submitted to AIH exhibit a greater respiratory than sympathetic response. Thus, the observed changes in PNA and RSNA responses under volatile anesthesia may have a background in volatile agents interacting on neural networks in the chemoreflex loop, precisely with the brainstem neurons and more rostral parts of the central nervous system [29,49,50,51,52]. However, the present model does not allow for making clear conclusions, considering multiple sites of action of volatile anesthetics.

The experimental model used in this study does not allow for a true unanesthetized animal control group, and the potential impact of urethane anesthesia on the results of this study cannot be disregarded. However, urethane anesthesia has been utilized in a multitude of previous studies [53] using similar models, due to its minimal effect on respiration and sympathetic nervous activity and can be considered as relevant for comparison [54,55]. It is well known that volatile anesthetics significantly interact with the central neuronal regions that regulate respiratory and sympathetic functions [55], where sevoflurane and isoflurane strongly enhance the GABAergic inhibition and reduce the excitability of glutamatergic neurons [28,29,30]. On the contrary, urethane does not seem to have a specific and predominant target for exerting its action and is preferentially used in experiments where the preservation of the reflexes is essential, as the observations closely mirror those made in conscious animals [53,54,55,56,57]. Previous reports indicate that both PNA and RSNA responses to chemoreflex activation may depend not only on the strength and type of the intermittent stimulus but also on the reoxygenation pattern [41,58,59,60]. It has been demonstrated that hyperoxia has minimal effects on the magnitude of PNA response but can significantly attenuate the sympathetic response to acute airway obstruction in rats [59]. The present study performed the AIH protocol in a hyperoxic background. A significant increase in RSNA from the baseline was observed only during the first hypoxic episode in the urethane-anesthetized group. On the contrary, in groups anesthetized with volatile anesthetics, the RSNA response was blunted throughout the AIH protocol. The applied hyperoxia (50% O_2_) during recovery periods may have influenced the adaptation of the sympathetic response to subsequent hypoxic challenges in the urethane group, while the initial sympathetic response to hypoxia in the volatile anesthetic groups may have been insufficient to evoke a similar pattern.

Additionally, it is important to consider the impact of volatile agents on MAP, considering this study used monoanesthesia with sevoflurane or isoflurane, which may reduce the MAP [61]. Moreover, acute hypoxia can lead to a decrease in MAP through the direct effect on vasculature through vasodilatation [62,63]. Thus, the combined effect of volatile anesthetics and powerful stimuli, such as severe hypoxia, on MAP might raise concern that the observed reduction in blood pressure might have affected both respiratory motor output and sympathetic outflow. This is in line with our previous studies that revealed that exposure to the same AIH protocol could lead to significant decreases in MAP during hypoxic episodes [23,44,60], as a consequence of the synergistic effect of the hypoxic stimulus and anesthesia itself, but it is difficult to separately account for the relative contribution from each side to MAP [61,62]. Both hypoxia and anesthesia may activate the baroreflex to increase the sympathetic outflow to target organs. However, the precise site of the anesthetic action is still unclear, due to a possibility of decreased sensitivity of aortic baroreceptors but also of a depression in the efferent pathway of the baroreflex at the level of sympathetic ganglia or the central nervous system [64]. In this study, MAP tended to decrease more profoundly during exposure to brief episodes of severe hypoxia in the urethane and isoflurane groups. However, MAP recovered to baseline values following exposure to AIH in all studied groups.

The limitation of this study is an uncertain and only indirect clinical impact since the conditions applied in the present protocol are rarely seen in real-life anesthesia. However, considering the complexity of the pathophysiology of OSA and its heterogeneity, an ideal research model regrettably does not exist. A better understanding of OSA may be achieved only through integration of all experimental approaches in humans or animal models. It is therefore evident that a more comprehensive exploration of the physiological mechanisms driving the AIH response across various physiological and clinical manifestations is essential. Basic research in animal models of intermittent hypoxia, as a well-established model used to study obstructive sleep apnea, provided valuable insights into the significance of blood gas alterations in systemic and end-organ changes in sleep apnea and proved to be useful in exploring the underlying mechanisms of pathophysiological changes in a range of medical conditions and disorders/diseases. Presently, there is an increasing number of OSA patients undergoing procedures under general anesthesia. They are not only susceptible to respiratory perioperative complications but also cardiovascular ones, such as arrhythmias, hypertension and cardiac dysfunction. Therefore, we believe that studies such as the present one are welcome, especially since the literature on this topic is rather scarce.

In conclusion, this study, for the first time, investigated the effects of monoanesthesia with volatile agents during brief, repetitive episodes of severe hypoxia while concomitantly recording the renal sympathetic and phrenic nerve activity. At equivalent MACs, the volatile anesthetics, isoflurane and sevoflurane, blunted the RSNA response. In addition, the PNA response to acute intermittent hypoxia was preserved, indicating that the respiratory system might be more robust than the sympathetic system response during exposure to acute intermittent hypoxia.

## Figures and Tables

**Figure 1 biomedicines-12-00910-f001:**
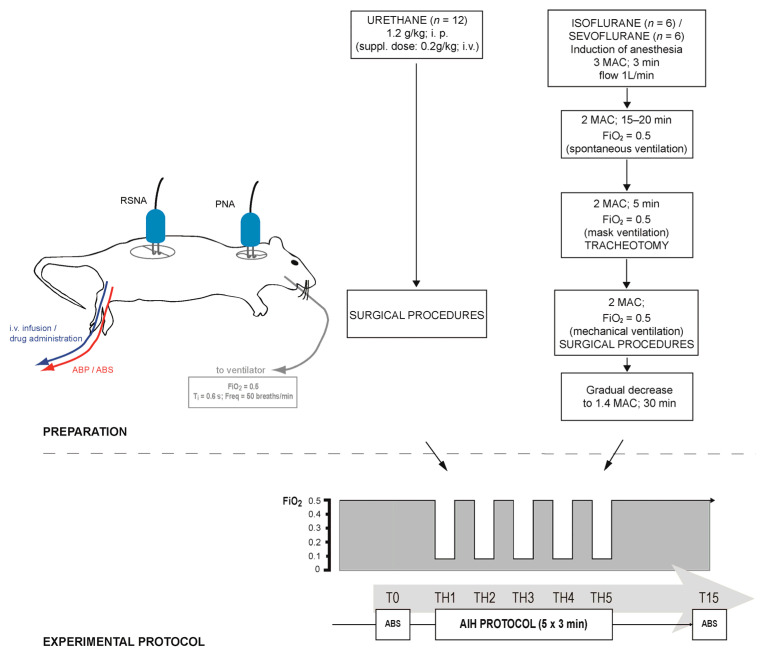
Outline of the experimental preparation and protocol in urethane (*n* = 12), isoflurane (*n* = 6) and sevoflurane (*n* = 6) groups. The acute intermittent hypoxia (AIH) protocol consisted in delivering five 3 min-long hypoxic episodes (fraction of inspired oxygen; FiO_2_ = 0.09), separated by 3 min recovery intervals (FiO_2_ = 0.5) at measured experimental time points (T0: baseline; TH1 to TH5: five hypoxic episodes; T15: 15 min following the AIH protocol). ABP: arterial blood pressure; ABS: acid–base status; Freq: ventilator frequency; MAC: minimum alveolar concentration; PNA: phrenic nerve activity; RSNA: renal sympathetic nerve activity; T_i_: inspiratory time.

**Figure 2 biomedicines-12-00910-f002:**
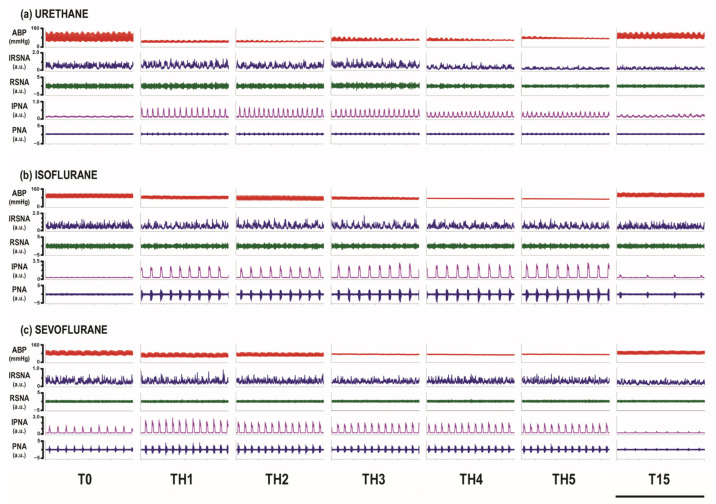
Compressed neurograms of the renal sympathetic nerve activity (RSNA) and phrenic nerve activity (PNA) at baseline (T0) during five hypoxic episodes (TH1 to TH5) and at 15 min following the acute intermittent hypoxia protocol (T15) in three studied groups: (**a**) urethane; (**b**) isoflurane; (**c**) sevoflurane. From top to bottom: arterial blood pressure (ABP) expressed in mmHg; integrated RSNA signal (IRSNA); raw RSNA signal; integrated PNA signal (IPNA) and raw PNA signal, all expressed in arbitrary units (a.u.). Scale bar represents 20 s.

**Figure 3 biomedicines-12-00910-f003:**
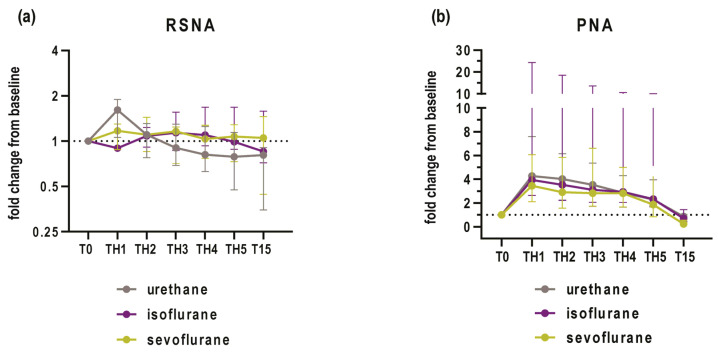
Fold changes in activities of (**a**) renal sympathetic nerve (RSNA) and (**b**) phrenic nerve (PNA) plotted as median and interquartile ranges at baseline (T0) during 5 hypoxic episodes (TH1 to TH5) and at 15 min following the acute intermittent hypoxia protocol (T15) in urethane, isoflurane and sevoflurane groups.

**Figure 4 biomedicines-12-00910-f004:**
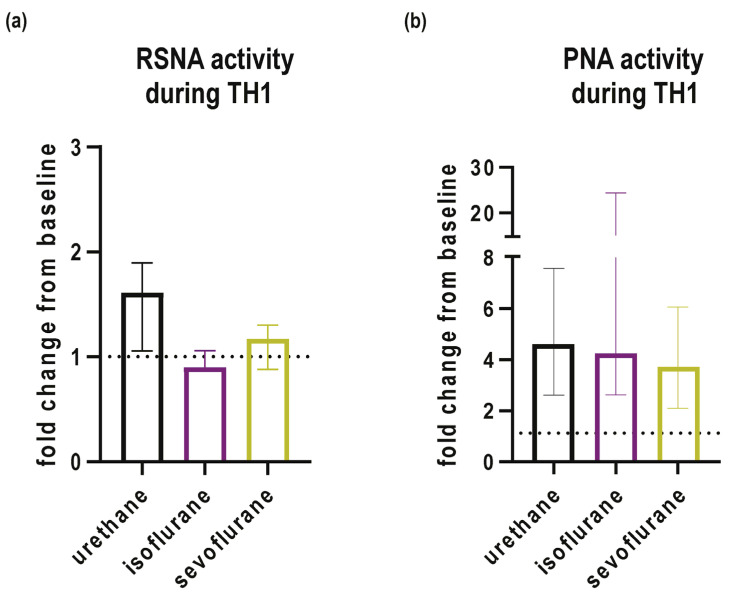
Fold changes in activities of (**a**) renal sympathetic nerve (RSNA) and (**b**) phrenic nerve (PNA) plotted as median and interquartile ranges during the first hypoxic episode (TH1) in urethane, isoflurane and sevoflurane groups. Dashed line represents baseline activity.

**Figure 5 biomedicines-12-00910-f005:**
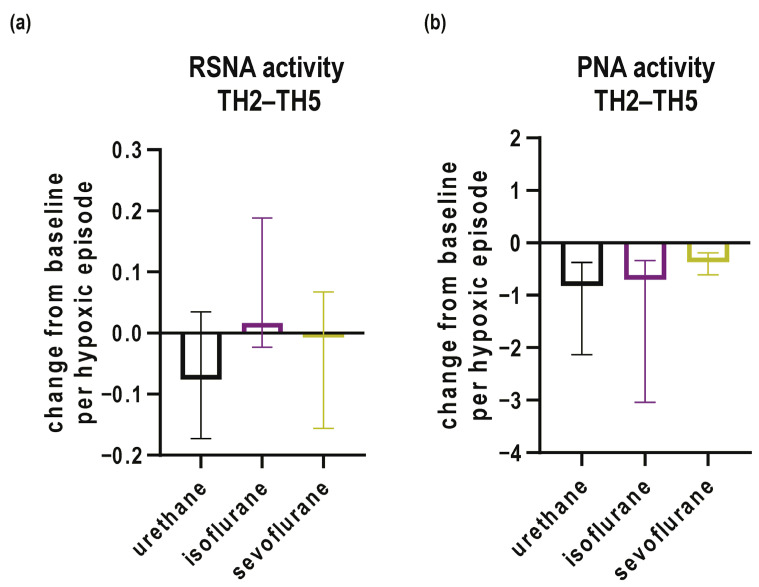
Changes from baseline in (**a**) renal sympathetic (RSNA) and (**b**) phrenic nerve activity (PNA) during 2nd to 5th episodes of hypoxia (TH2 to TH5) in urethane, isoflurane and sevoflurane groups. Data are expressed as changes in nerve activity per hypoxic episode and plotted as median and interquartile ranges.

**Table 1 biomedicines-12-00910-t001:** The values of mean arterial pressure (mmHg) in three studied groups at baseline (T0), during five hypoxic episodes (TH1 to TH5) and during recovery at 15 min (T15) following the exposure to acute intermittent hypoxia (AIH) protocol.

	Baseline	AIH Protocol					Recovery
Group	T0	TH1	TH2	TH3	TH4	TH5	T15
Urethane	98.5 ± 3.7	81.8 ± 6.8 ^†^	76.2 ± 6.4 ^†^	77.2 ± 5.2 *^,†^	72.9 ± 5.1 *^,†^	72.9 ± 5.6 *^,†^	106.2 ± 4.7
Isoflurane	82.1 ± 5.0 ^‡^	62.3 ± 8.5	62.1 ± 7.7	61.5 ± 6.1 *^,†^	60.5 ± 6.4 *^,†^	61.7 ± 5.9	87.4 ± 6.4
Sevoflurane	90.5 ± 2.6	73.6 ± 8.9	76.5 ± 8.8	64.7 ± 10.0	69.1 ± 7.6	72.9 ± 9.2	89.1 ± 4.1

Data are expressed as mean ± standard error of the mean. * Significantly different from baseline (*p* < 0.05; repeated measures ANOVA within group). ^†^ Significantly different from T15 (*p* < 0.05; repeated measures ANOVA within group). ^‡^ Significantly different from the urethane group (*p* < 0.05; one-way ANOVA between groups).

**Table 2 biomedicines-12-00910-t002:** Acid–base status at baseline (T0) and at 15 min (T15) following exposure to acute intermittent hypoxia protocol in isoflurane-, sevoflurane- and urethane-anesthetized groups.

Group	pH		PaCO_2_		PaO_2_	
	T0	T15	T0	T15	T0	T15
Urethane	7.300 ± 0.013	7.283 ± 0.015	52.0 ± 1.7	48.3 ± 2.0 *	263.9 ± 5.9	261.6 ± 6.2
Isoflurane	7.301 ± 0.022	7.276 ± 0.025	51.4 ± 1.5	50.3 ± 1.4	277.7 ± 8.9	281.3 ± 3.6
Sevoflurane	7.357 ± 0.029	7.356 ± 0.021	51.8 ± 1.7	49.5 ± 2.6	267.4 ± 12.3	265.8 ± 13.4

Data are expressed as mean ± standard error of the mean. PaCO_2_—partial pressure of carbon dioxide in arterial blood; PaO_2_—partial pressure of oxygen in arterial blood; both expressed in mmHg. * Significantly different from baseline (T0) (*p* = 0.045, repeated measures ANOVA within group).

## Data Availability

The data are contained within the article and the recordings and raw datasets supporting the conclusions of this study will be made available by the corresponding author on request.

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
