# Peer review of "The Effects of Volatile Anesthetics on Renal Sympathetic and Phrenic Nerve Activity during Acute Intermittent Hypoxia in Rats"

_biomedicines, 2024, doi:10.3390/biomedicines12040910_

Round 1

Reviewer 1 Report

Comments and Suggestions for Authors

Thank you for êrmitting me to review this manuscript

In this animal study the authors compared  the AIH in 3 groups of rats using sevoflurane , isoflurane , and urethane 

They conclude that volatile anesthetics blunt the AIH 

The authors should explain why blunting AIH is an important clinical issue and how these finding can be useful  in human medicine

  Table 1 need better formating 

Figure 1 the graph need better resolution 

The effect of volatile anesthesia on baroreflex  and interaaction with hypoxia should be better  discussed

conclusion:   it is difficult to confirm that volatile anesthetics blunted the response as there is no control group (I guess it would not be possible to have ethical approval) ,therefore the quthors should use a more cautious approach and discuss the shorthcoming of the study  

Reviewer 2 Report

Comments and Suggestions for Authors

This study is significant because it addresses the differential modulation of the sympathetic and respiratory systems under volatile anesthesia, a topic with implications for clinical anesthesia practices and understanding the pathophysiology of sleep-related breathing disorders.

This manuscript is well-designed and well-written.

  1. However, i have some major concerns to address:
  1. -The study involves a relatively small sample size, particularly for the experimental groups (n=6 for both sevoflurane and isoflurane groups). Was sample size estimated and how? Please add info in the manuscript.
  2.  
  3. - The control group is anesthetized with urethane, which is known to have different pharmacological effects compared to volatile anesthetics. The authors should discuss the rationale for choosing urethane as a control and its potential impact on the study's conclusions.
  4.  
  5. -The study focuses on the acute effects of volatile anesthetics during AIH episodes. It would be beneficial to include a discussion or future research directions on the long-term implications of these findings, especially considering the chronic nature of conditions like OSA.
  6.  
  7. -Authors should discuss the limitations of the rat model in the context of human anesthesia and OSA treatment.

- I found several english errors throughout the manuscript. Please have a deep language revision.

Comments on the Quality of English Language

Moderate revision required. 

Round 2

Reviewer 1 Report

Comments and Suggestions for Authors

The authors have adequately responded to my queries 

Reviewer 2 Report

Comments and Suggestions for Authors

Authors replied to my comments in a satisfactorily way. Therefore, this paper is ok to accept now.